# Determinants of knowledge of pregnancy danger signs in Indonesia

**Ratna Dwi Wulandari**[1]*, **Agung Dwi Laksono**[2]

**1** Faculty of Public Health, Universitas Airlangga, Surabaya, Indonesia, **2** National Institute of Health Research and Development, Indonesia Ministry of Health, Jakarta, Indonesia

* ratna-d-w@fkm.unair.ac.id

## Abstract

### Introduction

The maternal mortality rate in Indonesia is still quite high. It requires good knowledge for early prevention. The study aimed to analyze the determinants of knowledge of the pregnancy danger signs in Indonesia.

### Methods

The samples used were 85,832 women of childbearing age (15–49 years old). The variables included understanding of danger signs of pregnancy, types of residence, age, education, employment, marital status, wealth, parity, the autonomy of health, current pregnancy status, and media exposure. The determinant was pointed out by using binary logistic regression.

### Results

Urban women were 1.124 times more likely to understand the pregnancy danger signs of than rural women. Older women could identify pregnancy danger signs better than those aged 15–19 years. The more educated a woman is, the higher knowledge of the pregnancy danger signs she has. Married women or those who live with their partner were at 1.914 times likely to identify the pregnancy danger signs than unmarried ones or those who have never been in a relationship. If the wealth status gets higher, knowledge of the pregnancy danger signs will be better too. Grande multiparous women were at 0.815 times more likely to understand the pregnancy danger signs than primiparous. Women with the autonomy of health had 1.053 times chances to identify the pregnancy danger signs than those without autonomy. Women who were currently pregnant had 1.229 times better understanding of the pregnancy danger signs than women who were not currently pregnant. Media exposure had a good effect on women's understanding of the pregnancy danger signs.

**Data Availability Statement:** Data cannot be shared publicly because of the data are owned by a third party and authors do not have permission to share the data. The 2017 IDHS data set name requested from the ICF ('data set of childbearing age women') are available from the ICF (contact via

https://dhsprogram.com/data/new-user-registration.cfm) for researchers who meet the criteria for access to confidential data.

**Funding:** The author(s) received no specific funding for this work.

**Competing interests:** The authors have declared that no competing interests exist.

## Conclusion

All variables tested were the determinants of knowledge of the pregnancy danger signs in Indonesia. These include residence, age, education, employment, marital status, wealth, parity, the autonomy of health, current pregnancy status, and media exposure.

## Introduction

Globally, at least one woman dies every minute during pregnancy and childbirth [1]. Deaths due to pregnancy complications and vaginal birth can be easily prevented [2][3]. Prevention can be done by increasing knowledge of the pregnancy danger signs, which has a strong correlation with early detection of pregnancy risks. Women who know the pregnancy danger signs are at 6.657 times more likely than those who do not understand about early detection of pregnancy risks [4]. Knowledge of pregnancy danger signs has a strong correlation with antenatal care [3]. Women who can identify the pregnancy danger signs are 3.470 times more likely to participate in antenatal care [5]. It is an evidence that knowledge of pregnancy danger signs as well as Maternal Mortality Rate (MMR).

The Indonesian Government targeted several indicators for the development of health and nutrition status in 2019. First, the Maternal Mortality Rate (MMR) was 306 per 100,000 live births. Second, the Infant Mortality Rate (IMR) was targeted to reach 24 per 1,000 live births. Third, the prevalence of malnutrition among children under five years was 17 per 100,000 live births. Fourth, the prevalence of stunting among children under two years was 28 per 100,000 population [6].

Based on the latest data released by the Data and Information Center in 2016, the Indonesian Ministry of Health reported that at this time MMR decreased. The 2010 Population Census showed there were 346 maternal mortalities per 100,000 live births, and then the number of maternal deaths dropped to 305 per 100,000 live births based on 2015 Inter-Census Population Survey (SUPAS) data. However, this figure has still not reached the MDG target in 2015 of 102 per 100,000 live births [7].

While based on the SDG's target, Indonesia is demanded to achieve even higher. There are 3 main targets, which are likely to reduce the MMR below 70 deaths per 100,000 live births, the number of neonatal mortalities of 12 per 1000 live births, and the number of death rate among under-five-year children by 25 per 1,000 live births [8].

Compared to other countries, the MMR in Indonesia had a higher rate. The MMR recorded in Indonesia was 9 times higher than in Malaysia, 5 times higher compared to in Vietnam, and almost 2 times higher than in Cambodia. The World Health Organization estimated that a significant disparity of the MMR occurred between developed and developing countries. The MMR in developed countries was approximately in the range of 12 per 100,000 live births, while in developing countries it was around at 239 per 100,000 live births [9][10].

Indonesia has to put more efforts to reduce the MMR. Massive community engagement is required, especially among women, to understand the pregnancy danger signs. It can raise women's awareness as to anticipate any dangers [11][4]. Women who perceive risks can immediately consult health workers.

Promoting right pregnancy danger signs need to be widely done in Indonesia. It is necessary because Indonesia has hundreds of ethnic groups with diverse cultures, some of which have conservative knowledge of pregnancies that are contradictory to modern midwifery

knowledge [12][13]. Not only general public but also health workers who have received modern medical education still have conservative knowledge about pregnancy and childbirth [14].

This situation has raised an interesting question to analyze the determinants of knowledge of pregnancy danger signs in Indonesia. The results of this study may be clear and directed guidelines for policymakers in determining the policy objectives of disseminating the pregnancy danger signs to reduce the MMR in Indonesia.

## Methods

### Data source

The secondary data from the 2017 Indonesian Demographic Data Survey (IDHS) were used for analysis. The IDHS was part of the Demographic and Health Survey (DHS) series. The DHS was internationally conducted by the Inner City Fund (ICF). The sampling method in the IDHS used stratification and multistage random sampling. In this study, the units of analysis were 85,832 women in childbearing age (15–49 years).

### Procedure

The 2017 IDHS has passed the ethical test from the National Ethics Committee. The respondents' identities have all been deleted from the dataset. Respondents have provided written approval for their involvement in the study. The researchers obtained the consent of data utilization from ICF International by applying on their website: https://dhsprogram.com/data/new-user-registration.cfm.

### Data analysis

Knowledge of pregnancy danger signs was defined as knowledge of dangers of prolonged labor, vaginal bleeding, fever, convulsions, breech position, swollen limbs, faint, breathlessness, tiredness, and others. Abilities to identify danger signs of pregnancy are divided into 2 categories; "do not know" and "know". Respondents were considered "know" when they claimed to know all pregnancy danger signs.

Independent variables involved in the analysis include types of residence, age groups, education level, employment status, marital status, wealth status, parity, autonomy of health, current pregnant status, frequency of reading newspaper/magazine, frequency of listening radio, and frequency of watching television. Types of residence are divided into 2 categories, which are "urban" and "rural". Age group is divided into 7 categories with 5-year interval. Education level consists of 4 categories, such as "no education", "primary education", "secondary education" and "higher education". Employment status is divided into 2 categories, such as "no employment" and "employment".

Marital status is divided into 3 categories, for instance, "never in a union", "married or living with partners", and "widowed or divorced". Wealth status is determined based on the wealth index calculation. Wealth index is a composite measure of a household's cumulative living standard. Wealth index was calculated by listing household ownership of selected assets, such as televisions and bicycles, materials used for housing construction, and types of water access and sanitation facilities. There are five categories of wealth index, such as "the poorest", "poorer", "middle", "richer", and "the richest".

Parity, in addition, is the number of children ever born alive. Parity is divided into 3 categories, for instances, "primiparous ($\leq$ 1)", "multiparous (2–4)", and "grand multiparous ($>$4)". Autonomy of health is the independence to determine the needs of health services. Autonomy of health has 2 categories, which are "not having autonomy" and "having autonomy". Current

pregnancy is the current state of pregnancy status during the interview, which has 2 categories, "not pregnant" and "pregnant".

The last variable group is media exposure, such as newspaper/magazine, radio, and television. Intensity of media exposure is categorized into "not at all", "less than once a week", and "at least once a week".

The collinearity test was used at an early stage to ensure no collinearity between variables. All variables involved in the analysis were dichotomous variables, and thus the chi-square test was used to determine whether there are significant differences in knowledge of pregnancy danger signs in Indonesia. In the final stage, the binary logistic regression was used because of the nature of the dependent variable. All statistical analyses were carried out in SPSS 22 software.

## Results

Table 1 figures out the results of the variable collinearity test as a predictor of knowledge of pregnancy danger signs in Indonesia. The collinearity test showed no collinearity between the dependent and independent variables.

The tolerance value of all variables as shown in Table 1 is greater than 0.10. While the VIF value for all variables is less than 10.00. Referring to the basis of multicollinearity test, it can be concluded that there was no multicollinearity in the regression model.

### Descriptive results

Table 2 displays descriptive statistics of knowledge of pregnancy danger signs in Indonesia. Table 2 informs that women who did not know about the pregnancy danger signs were dominated by those who lived in rural areas. While women who knew the pregnancy danger signs predominantly lived in urban areas. The senior age group (45–49 years old) are domineted by women who did not know the pregnancy danger signs. While those who claimed to identify the danger signs were mostly in the middle age group.

Table 2 informs those with no knowledge of the pregnancy danger signs were dominated by female graduates by level of primary school. While women with secondary education

**Table 1. Results for the co-linearity test of knowledge of the pregnancy danger signs in Indonesia (n = 85,832).**

| VARIABLES | COLLINEARITY STATISTICS | |
|---|---|---|
| | Tolerance | VIF |
| Type of place of residence | 0.758 | 1.319 |
| Age | 0.725 | 1.380 |
| Education level | 0.671 | 1.491 |
| Employment status | 0.944 | 1.059 |
| Marital status | 0.931 | 1.074 |
| Wealth status | 0.604 | 1.656 |
| Parity | 0.946 | 1.057 |
| The autonomy of Health | 0.734 | 1.362 |
| Curently pregnant | 0.975 | 1.025 |
| Frequency of reading newspaper/magazine | 0.741 | 1.349 |
| Frequency of listening to a radio | 0.873 | 1.145 |
| Frequency of watching television | 0.900 | 1.112 |

*Dependent Variable: Know of the pregnancy danger signs

**Table 2. Descriptive statistic of knowledge of the danger signs of pregnancy in Indonesia (n = 85,832).**

| CHARACTERISTICS | The Knowledge of the Pregnancy Danger Signs | | | | P |
|---|---|---|---|---|---|
| | Do not know | | Know | | |
| | n | % | n | % | |
| **Type of place of residence** | | | | | \*\*\*< 0.001 |
| - Urban | 14877 | 39.9% | 26308 | 54.1% | |
| - Rural (ref.) | 22370 | 60.1% | 22277 | 45.9% | |
| **Age groups** | | | | | \*\*\*< 0.001 |
| - 15–19 (ref.) | 233 | 0.6% | 211 | 0.4% | |
| - 20–24 | 1327 | 3.6% | 2021 | 4.2% | |
| - 25–29 | 3094 | 8.3% | 5246 | 10.8% | |
| - 30–34 | 5489 | 14.7% | 8892 | 18.3% | |
| - 35–39 | 8045 | 21.6% | 11240 | 23.1% | |
| - 40–44 | 9147 | 24.6% | 11246 | 23.1% | |
| - 45–49 | 9912 | 26.6% | 9729 | 20.0% | |
| **Education level** | | | | | \*\*\*< 0.001 |
| - No education (ref.) | 2161 | 5.8% | 737 | 1.5% | |
| - Primary | 18069 | 48.5% | 14327 | 29.5% | |
| - Secondary | 15287 | 41.0% | 25839 | 53.2% | |
| - Higher | 1730 | 4.6% | 7682 | 15.8% | |
| **Employment status** | | | | | \*\*0.005 |
| - No Employed | 14647 | 39.3% | 19569 | 40.3% | |
| - Employed | 22600 | 60.7% | 29016 | 59.7% | |
| **Marital status** | | | | | \*\*\*< 0.001 |
| - Never in union | 33 | 0.1% | 21 | 0.0% | |
| - Married/living with partner | 34422 | 92.4% | 45967 | 94.6% | |
| - Widowed/divorced | 2792 | 7.5% | 2597 | 5.3% | |
| **Wealth status** | | | | | \*\*\*< 0.001 |
| - Poorest (ref.) | 13713 | 36.8% | 10093 | 20.8% | |
| - Poorer | 7974 | 21.4% | 8783 | 18.1% | |
| - Middle | 6449 | 17.3% | 9065 | 18.7% | |
| - Richer | 5222 | 14.0% | 9901 | 20.4% | |
| - Richest | 3889 | 10.4% | 10743 | 22.1% | |
| **Parity** | | | | | \*\*\*< 0.001 |
| - Primiparous (ref.) | 3139 | 8.4% | 5493 | 11.3% | |
| - Multiparous | 23984 | 64.4% | 35149 | 72.3% | |
| - Grandemultiparous | 10124 | 27.2% | 7943 | 16.3% | |
| **The autonomy of Health** | | | | | \*\*\*< 0.001 |
| - No | 22924 | 61.5% | 27916 | 57.5% | |
| - Yes | 14323 | 38.5% | 20669 | 42.5% | |
| **Currently pregnant** | | | | | \*\*\*< 0.001 |
| - No | 36352 | 97.6% | 46938 | 96.6% | |
| - Yes | 895 | 2.4% | 1647 | 3.4% | |
| **Frequency of reading newspaper/magazine** | | | | | \*\*\*< 0.001 |
| - Not at all (ref.) | 26740 | 71.8% | 26407 | 54.4% | |
| - Less than once a week | 8861 | 23.8% | 16710 | 34.4% | |
| - At least once a week | 1646 | 4.4% | 5468 | 11.3% | |
| **Frequency of listening to a radio** | | | | | \*\*\*< 0.001 |
| - Not at all (ref.) | 25721 | 69.1% | 28178 | 58.0% | |

(*Continued*)

**Table 2.** (Continued)

| CHARACTERISTICS | The Knowledge of the Pregnancy Danger Signs | | | | P |
| | Do not know | | Know | | |
| | n | % | n | % | |
| - Less than once a week | 8380 | 22.5% | 14298 | 29.4% | |
| - At least once a week | 3146 | 8.4% | 6109 | 12.6% | |
| **Frequency of watching television** | | | | | ***< 0.001 |
| - Not at all (ref.) | 3293 | 8.8% | 1803 | 3.7% | |
| - Less than once a week | 5574 | 15.0% | 5666 | 11.7% | |
| - At least once a week | 28380 | 76.2% | 41116 | 84.6% | |

* p < 0.05

** p < 0.01

***p < 0.001.

predominantly know the pregnancy danger signs. In terms of employment status, both categories were dominated by employed women.

Table 2 shows that the poorest women mostly did not know the pregnancy danger signs. While women who knew the pregnancy danger signs had a more equitable distribution of wealth status.

In terms of parity variable, both categories were dominated by multiparous women. Most the respondents have their autonomy of health. In current pregnancy status, women who were not pregnant dominated the groups.

In addition, most of the respondents have no exposure to newspaper/magazine and radio. While the respondents mostly claimed to watch television at least once a week.

## Multivariate regression analysis

The results of a binary logistic regression test on knowledge of pregnancy danger signs in Indonesia are illustrated in Table 3. This statistical test could determine the determinants of knowledge of pregnancy danger signs in Indonesia. As a reference, the chosen category was "do not know the pregnancy danger signs".

Table 3 depicts that women who lived in urban areas have 1.124 times chance to know the pregnancy danger signs than women in rural areas (OR 1.124; 95% CI 1.088–1.161). Older age groups have a better chance of knowing the pregnancy danger signs than those in the age of 15–19 years as reference. Only the groups aged 45–49 years have no difference with the reference age group.

Results show the more educated a woman is, the higher the likelihood of knowing the pregnancy danger signs is. Women with higher education were 4.902 times more likely to identify pregnancy danger signs than women with no education (OR 4.902; 95% CI 4.404–5.457). Employed women were 0.963 times more likely to spot pregnancy danger signs than unemployed women (OR 0.963; 95% CI 0.934–0.992).

While women who married or lived with partner had 1.914 times possibilities to identify pregnancy danger signs than women who have never been in relationship (OR 1.078–3.397). The better the wealth status of a woman is, the more knowledge of pregnancy danger signs is. The richest woman group had 1.758 times chances to have better knowledge than the poorest woman group (OR 1.758; 95% CI 1.662–1.859).

Besides, women with health insurance had 1.155 times chance for better knowledge of pregnancy danger signs than those without health insurance (OR 1.155; 95% CI 1.121–1.190). The possibility to have knowledge of pregnancy danger sings are 0.815 times for grand multiparous

**Table 3. Binary logistic regression of knowledge of the pregnancy danger signs in Indonesia (n = 85,832).**

| PREDICTOR | The Knowledge of The Pregnancy Danger Signs | | | |
|---|---|---|---|---|
| | Sig. | OR | Lower Bound | Upper Bound |
| Type of place of residence: Urban | \*\*\*< 0.001 | 1.124 | 1.088 | 1.161 |
| Type of place of residence: Rural | - | - | - | - |
| Age group: 15–19 | - | - | - | - |
| Age group: 20–24 | \*\*\*< 0.001 | 1.545 | 1.259 | 1.897 |
| Age group: 25–29 | \*\*\*< 0.001 | 1.607 | 1.315 | 1.964 |
| Age group: 30–34 | \*\*\*< 0.001 | 1.576 | 1.290 | 1.925 |
| Age group: 35–39 | \*\*\*< 0.001 | 1.453 | 1.190 | 1.776 |
| Age group: 40–44 | \*\*0.001 | 1.398 | 1.144 | 1.709 |
| Age group: 45–49 | 0.069 | 1.205 | .986 | 1.474 |
| Education level: No Education | - | - | - | - |
| Education level: Primary | \*\*\*< 0.001 | 1.646 | 1.506 | 1.800 |
| Education level: Secondary | \*\*\*< 0.001 | 2.582 | 2.357 | 2.828 |
| Education level: Higher | \*\*\*< 0.001 | 4.902 | 4.404 | 5.457 |
| Employment status: Not employed | - | - | - | - |
| Employment status: Employed | \*0.013 | 0.963 | 0.934 | 0.992 |
| Marital status: Never in union | - | - | - | - |
| Marital status: Married/living with partner | \*0.027 | 1.914 | 1.078 | 3.397 |
| Marital status: Widowed/divorced | 0.143 | 1.539 | .865 | 2.737 |
| Wealth status: Poorest | - | - | - | - |
| Wealth status: Poorer | \*\*\*< 0.001 | 1.174 | 1.124 | 1.225 |
| Wealth status: Middle | \*\*\*< 0.001 | 1.337 | 1.277 | 1.401 |
| Wealth status: Richer | \*\*\*< 0.001 | 1.581 | 1.504 | 1.661 |
| Wealth status: Richest | \*\*\*< 0.001 | 1.758 | 1.662 | 1.859 |
| Parity: Primiparous | - | - | - | - |
| Parity: Multiparous | 0.571 | 0.984 | 0.931 | 1.040 |
| Parity: Grande multiparous | \*\*\*< 0.001 | 0.815 | 0.763 | 0.870 |
| The autonomy of health: No | - | - | - | - |
| The autonomy of health: yes | \*\*0.001 | 1.053 | 1.022 | 1.085 |
| Currently pregnant: No | - | - | - | - |
| Currently pregnant: Yes | \*\*\*< 0.001 | 1.229 | 1.125 | 1.341 |
| Freq. of reading news/magazine: Not at all (ref.) | - | - | - | - |
| Freq. of reading news/magazine: Less than once a week | \*\*\*< 0.001 | 1.288 | 1.243 | 1.335 |
| Freq. of reading news/magazine: At least once a week | \*\*\*< 0.001 | 1.510 | 1.416 | 1.610 |
| Freq. of listening radio: Not at all (ref.) | - | - | - | - |
| Freq. of listening radio: Less than once a week | \*\*\*< 0.001 | 1.153 | 1.112 | 1.195 |
| Freq. of listening radio: At least once a week | \*\*\*< 0.001 | 1.234 | 1.174 | 1.297 |
| Freq. of watching television: Not at all (ref.) | - | - | - | - |
| Freq. of watching television: Less than once a week | \*\*\*< 0.001 | 1.204 | 1.119 | 1.294 |
| Freq. of watching television: At least once a week | \*\*\*< 0.001 | 1.344 | 1.258 | 1.435 |

\*p < 0.05

\*\* p < 0.01

\*\*\*p < 0.001.

women (OR 0.815; 95% CI 0.763–0.870). Birth experience does not automatically improve the respondents' knowledge of pregnancy danger signs.

Autonomy of health gave women 1.053 times chances to spot the pregnancy danger signs (OR 1.053; 95% CI 1.022–1.085). Women who were currently pregnant were 1.229 times more likely to know the pregnancy danger signs than women who were not currently pregnant (OR 1.229; 95% CI 1.125–1.341).

Frequent media exposure has a good impact on improving knowledge of the pregnancy danger signs. Women who read newspapers/magazines at least once a week had 1.510 times chances to identify the pregnancy danger signs than those who did not read newspapers/magazines (OR 1.510; 95% CI 1.416–1.610). Whereas, women who listened to radio at least once a week were 1.234 times more likely to have better knowledge (OR 1.234; 95% CI 1.174–1.297). The last point higlights that women who watched television at least once a week had 1.344 times chances to identify the pregnancy danger signs (OR 1.344; 95% CI 1.258–1.435).

## Discussion

The findings reported that as many as 56.66% of pregnant women in Indonesia claimed to have knowledge of pregnancy danger signs. The percentage of pregnant woment with knowledge of pregnancy danger signs is higher compared to that in Ethiopia at 40.0% [3] and Nigeria at 42.4% [15]. A study in Papua New Guinea and Tanzania, however, found even much higher percentage. Research in Papua New Guinea informed that 60.2% of women could mention at least one of the pregnancy danger signs [16]. While in Tanzania only 57.8% of women could mention at least 1–3 the pregnancy danger signs [17].

The results also found that women in urban areas were more likely to identify the pregnancy danger signs than women in rural areas. These findings support several previous studies that focused on discussing about disparities between urban-rural areas in Indonesia. The development in health sector in Indonesia is indeed more massive in urban areas [18][19]. Similar research findings were also found in Somali and Northern Ethiopia [20][21].

Older age groups had a better chance of knowing the pregnancy danger signs than those aged 15–19 years as a reference. Only the age group of 45–49 years had no difference with the reference age group. The youngest age group tend to have less experience, and the oldest groups had a more conservative view. A systematic review and meta-analysis of women's knowledge of the obstetric danger signs in Ethiopia found similar results. Age is one of the variables that influences knowledge of the pregnancy danger signs, in addition to several other demographic characteristics [22].

Education also affects knowledge of the pregnancy danger signs. Women with higher education had more chances to identify the pregnancy danger signs. This study higlights the same findings as the research in Papua New Guinea, Ethiopia, and Tanzania. These studies discovered that women with secondary education had a better chance of knowing of the pregnancy danger signs than women with no education and primary education [16][23][17]. The results also found that employment also influenced knowledge of the pregnancy danger signs in Indonesian pregnant women. Several studies found similar results in Malaysia, Tanzania and Ethiopia [23][24][25].

Married/living with partner women had 1.914 times possibilities to have greater knowledge of the pregnancy danger signs compared to women who have never been in relationship. In Indonesia, women who are in relationship but are pregnant are considered as a social disgrace. This condition encourages women socially conceal themselves from society [26].

The better the wealth status of a woman is, the higher the possibility to have knowledge of the pregnancy danger signs is. Like education level, several other studies have also found that wealth status was proven to be positively related to knowledge of the pregnancy danger signs [24][25][26][27].

Grand multiparous women were less likely to have better knowledge than primiparous women, but birth experience did not automatically improve their knowledge. This study shows different results from several studies in India and Ethiopia. These studies found that multiparous women had a better chance to identify the pregnancy danger signs and obstetric complications [21][28][29].

Women who had autonomy of health had a better chance to know the pregnancy danger signs than women without autonomy. A meta-analysis of 12 studies in Ethiopia found the same results. Autonomy will increase knowledge of the pregnancy danger signs [22]. This present study discovered that women who were currently pregnant were more likely to know the pregnancy danger signs. Pregnancy experience will increase awareness and curiosity about their condition [21][30].

The analysis found that women who get more exposed to media had better knowledge. This finding is in line with the findings of other previous studies, which confirm that media exposure is the best tool for increasing knowledge [4]. Meanwhile, another study in Indonesia about the effects of the Maternal and Child Health Handbook on improving knowledge of the pregnancy danger signs found contradictory results. It was concluded that the Maternal and Child Health Handbook could not improve knowledge of the pregnancy danger signs [31].

In general, better knowledge of the pregnancy danger signs is one of the determinants of early pregnancy detection [4]. Vigilance against the pregnancy danger signs is one of the right strategies to reduce the maternal mortality [11]. Besides health workers [32], mass media is the most popular source of searching more information about the pregnancy danger signs [30].

## Conclusions

In conclusion, all variables tested were the determinants of knowledge of the pregnancy danger signs in Indonesia. These variables were types of residence, age groups, education level, employment status, marital status, wealth status, health insurance, parity, autonomy of health, current pregnant status, frequency of reading newspaper/magazine, frequency of listening to the radio, and frequency of watching television.

The government has to formulate structured policies for the targets to expand the dissemination of knowledge of the pregnancy danger signs. This study recommends the government to focus on the research findings in relation to determine the policy targets.

## Author Contributions

**Conceptualization:** Ratna Dwi Wulandari, Agung Dwi Laksono.

**Data curation:** Ratna Dwi Wulandari, Agung Dwi Laksono.

**Formal analysis:** Ratna Dwi Wulandari, Agung Dwi Laksono.

**Investigation:** Ratna Dwi Wulandari, Agung Dwi Laksono.

**Methodology:** Ratna Dwi Wulandari, Agung Dwi Laksono.

**Project administration:** Agung Dwi Laksono.

**Resources:** Ratna Dwi Wulandari, Agung Dwi Laksono.

**Software:** Agung Dwi Laksono.

**Supervision:** Ratna Dwi Wulandari.

**Validation:** Ratna Dwi Wulandari.

**Writing – original draft:** Ratna Dwi Wulandari, Agung Dwi Laksono.

**Writing – review & editing:** Ratna Dwi Wulandari, Agung Dwi Laksono.

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
