## [Decision Letter · Decision Letter 0]

11 Mar 2020

PONE-D-20-02732

The Determinant of Knowledge of The Pregnancy Danger Signs in Indonesia

PLOS ONE

Dear Dr Wulandari,

Thank you for submitting your manuscript to PLOS ONE. After careful consideration, we feel that it has merit but does not fully meet PLOS ONE’s publication criteria as it currently stands. Therefore, we invite you to submit a revised version of the manuscript that addresses the points raised during the review process.

SPECIFIC ACADEMIC EDITOR COMMENTS: Your manuscript was handled by an expert reviewer in the field. Although interest was found in this study, there were several major comments that arose, which need addressing. These comments and concerns relate to the need for the English to be carefully proofed. Furthermore, the data sources used in this analyses should include more up-to-date studies. There are also manuscript-fundamentals that require attention whereby the introduction needs better rationale explaining the need to conduct this study and its novelty; the methods need more specifics about inclusion criteria; and the discussion should be supported by published studies and the current findings from this analysis.

We would appreciate receiving your revised manuscript by Apr 25 2020 11:59PM. To enhance the reproducibility of your results, we recommend that if applicable you deposit your laboratory protocols in protocols.io, where a protocol can be assigned its own identifier (DOI) such that it can be cited independently in the future. For instructions see: http://journals.plos.org/plosone/s/submission-guidelines#loc-laboratory-protocols

We look forward to receiving your revised manuscript.

Kind regards,

Frank T. Spradley

Academic Editor

PLOS ONE

Journal Requirements:

Reviewers' comments:

Reviewer's Responses to Questions

**Comments to the Author**

1. Is the manuscript technically sound, and do the data support the conclusions?

Reviewer #1: Partly

2. Has the statistical analysis been performed appropriately and rigorously? 

Reviewer #1: Yes

3. Have the authors made all data underlying the findings in their manuscript fully available?

Reviewer #1: No

4. Is the manuscript presented in an intelligible fashion and written in standard English?

Reviewer #1: No

5. Review Comments to the Author

Reviewer #1: Thank you for giving me the opportunity to review the manuscript entitled: "The determinant of knowledge of the pregnancy danger signs"

However, It is unfortunate that the journal did not provide me the article with line numbers track, that makes a bit difficult to give more details comments.

Please find below is my comments:

- An English Language editing will make the article more interesting to read as I found some words were not put in the correct way, e.g. page 3 – almost 2 times that of Cambodia… etc.

- The data source was from the 2017 IDHS data meaning the data was gathered could be a year before or more. A new information may exist.

- It is unclear the inclusion criteria for a woman childbearing age (15-49), as 15 -19 used as reference for some data, were any of those women aged 15 years old experience pregnancy?

- Were there any data regarding the MMR in pregnancy because the women had chronic disease?

- The discussion was not support with a strong evidence from previous studies that explain the knowledge of pregnancy danger signs have a significant correlation with MMR

- The background need to be added with some literature that explains the gap in the literature why is so important to do this research.

6. PLOS authors have the option to publish the peer review history of their article (what does this mean?). If published, this will include your full peer review and any attached files.

Reviewer #1: No

---

## [Author Response · Author response to Decision Letter 0]

18 Mar 2020

1) An English Language editing will make the article more interesting to read as I found some words were not put in the correct way, e.g. page 3 – almost 2 times that of Cambodia… etc.

Thank you for the advice. The revised manuscript was professionally edited.

2) The data source was from the 2017 IDHS data meaning the data was gathered could be a year before or more. A new information may exist.

The latest official release of MMR is not yet available by the Indonesian government.

3) It is unclear the inclusion criteria for a woman childbearing age (15-49), as 15 -19 used as reference for some data, were any of those women aged 15 years old experience pregnancy?

Based on the information in Table 2, there are 444 women aged 15-19 who experience pregnancy.

4) Were there any data regarding the MMR in pregnancy because the women had chronic disease?

No information is available for the data was mentioned.

5) The discussion was not support with a strong evidence from previous studies that explain the knowledge of pregnancy danger signs have a significant correlation with MMR

Globally, at least one woman dies every minute during pregnancy and childbirth [1]. Deaths due to pregnancy complications and vaginal birth can be easily prevented [2][3]. Prevention can be done by increasing knowledge of the pregnancy danger signs, which has a strong correlation with early detection of pregnancy risks. Women who know the pregnancy danger signs are at 6.657 times more likely than those who do not understand about early detection of pregnancy risks [4]. Knowledge of pregnancy danger signs has a strong correlation with antenatal care [3]. Women who can identify the pregnancy danger signs are 3.470 times more likely to participate in antenatal care [5]. It is an evidence that knowledge of pregnancy danger signs as well as Maternal Mortality Rate (MMR).

6) The background need to be added with some literature that explains the gap in the literature why is so important to do this research.

Globally, at least one woman dies every minute during pregnancy and childbirth [1]. Deaths due to pregnancy complications and vaginal birth can be easily prevented [2][3]. Prevention can be done by increasing knowledge of the pregnancy danger signs, which has a strong correlation with early detection of pregnancy risks. Women who know the pregnancy danger signs are at 6.657 times more likely than those who do not understand about early detection of pregnancy risks [4]. Knowledge of pregnancy danger signs has a strong correlation with antenatal care [3]. Women who can identify the pregnancy danger signs are 3.470 times more likely to participate in antenatal care [5]. It is an evidence that knowledge of pregnancy danger signs as well as Maternal Mortality Rate (MMR).

---

## [Decision Letter · Decision Letter 1]

17 Apr 2020

Determinant of Knowledge of Pregnancy Danger Signs in Indonesia

PONE-D-20-02732R1

Dear Dr. Wulandari,

We are pleased to inform you that your manuscript has been judged scientifically suitable for publication and will be formally accepted for publication once it complies with all outstanding technical requirements.

With kind regards,

Frank T. Spradley

Academic Editor

PLOS ONE

Reviewers' comments:

Reviewer's Responses to Questions

**Comments to the Author**

1. If the authors have adequately addressed your comments raised in a previous round of review and you feel that this manuscript is now acceptable for publication, you may indicate that here to bypass the “Comments to the Author” section, enter your conflict of interest statement in the “Confidential to Editor” section, and submit your "Accept" recommendation.

Reviewer #1: All comments have been addressed

2. Is the manuscript technically sound, and do the data support the conclusions?

Reviewer #1: Partly

3. Has the statistical analysis been performed appropriately and rigorously? 

Reviewer #1: I Don't Know

4. Have the authors made all data underlying the findings in their manuscript fully available?

Reviewer #1: Yes

5. Is the manuscript presented in an intelligible fashion and written in standard English?

Reviewer #1: Yes

6. Review Comments to the Author

Reviewer #1: The authors has addressed most of the comments in the revised manuscript. Although I found the discussion is not deep enough explained about the results or data compared to previous studies. It could be better presentation if the authors add more evidence in the previous studies.

7. PLOS authors have the option to publish the peer review history of their article (what does this mean?). If published, this will include your full peer review and any attached files.

Reviewer #1: No

---

## [Editor Report · Acceptance letter]

8 May 2020

PONE-D-20-02732R1 

Determinants of Knowledge of Pregnancy Danger Signs in Indonesia 

Dear Dr. Wulandari:

I am pleased to inform you that your manuscript has been deemed suitable for publication in PLOS ONE. Congratulations! Your manuscript is now with our production department. 

With kind regards,

on behalf of

Dr. Frank T. Spradley 

Academic Editor

PLOS ONE